# Tumor-Promoting Actions of HNRNP A1 in HCC Are Associated with Cell Cycle, Mitochondrial Dynamics, and Necroptosis

**DOI:** 10.3390/ijms231810209

**Published:** 2022-09-06

**Authors:** Biao Zhao, Xiaochen Lv, Xiaoqi Zhao, Subinuer Maimaitiaili, Yuheng Zhang, Ke Su, Hang Yu, Cheng Liu, Tong Qiao

**Affiliations:** 1Department of Vascular Surgery, Nanjing Drum Tower Hospital, The Affiliated Hospital of Nanjing University Medical School, Nanjing 210008, China; 2Nanjing Drum Tower Hospital Clinical College of Traditional Chinese and Western Medicine, Nanjing University of Chinese Medicine, Nanjing 210008, China

**Keywords:** HNRNP A1, HCC, cell senescence, cell cycle, P16^INK4^

## Abstract

Hepatocellular carcinoma (HCC) is one of the most frequent malignancies in the world. Although increasing evidence supports the role of heterogeneous ribonucleoprotein particle A1 (HNRNP A1) in tumor progression, the function of HNRNP A1 in HCC remains unclear. Here, we focused on the role of HNRNP A1 in the development of HCC. In this study, we found HNRNP A1 participates in many aspects of HCC, such as progression and prognosis. Our results showed that HNRNP A1 is upregulated in human HCC tissues and cell lines. High expression of HNRNP A1 can promote the proliferation, migration, and invasion in HCC cells and accelerate tumor progression in mice. Moreover, we found that HNRNP A1 prevents the senescence process of HCC cells. Knocking down of HNRNP A1 promotes the expression of P16^INK4^, which arrests the cell cycle and then induces the senescence phenotype in HCC cells. Furthermore, we found that HNRNP A1 regulated necroptosis and mitochondrial dynamics. In summary, our study indicates that HNRNP A1 promotes the development of HCC, which suggests a potential therapeutic target for HCC.

## 1. Introduction

Hepatocellular carcinoma (HCC) is the sixth-most-prevalent cancer in the world, and the fourth-most-relevant cause of cancer-related death following lung cancer, colorectal cancer, and stomach cancer [1]. Although some current treatments are available for hepatocellular carcinoma, such as surgery, radiofrequency ablation, trans-arterial chemoembolization, and systemic chemotherapy [2], the 5-year survival rate of HCC is only 18% [3], as the most majority of HCC patients are diagnosed at an advanced stage [4]. However, the mechanism of liver cancer has not been fully clarified. Recent studies have found that numerous factors can participate in the progression of hepatocellular carcinoma, such as cell cycle, necroptosis, and mitochondrial dynamics, and lead to the change of the cell fate of hepatocellular carcinoma [5,6,7].

Heterogeneous nuclear ribonucleoproteins (HNRNPs) are a kind of protein that can regulate the abundances and functions of RNA at various levels, which may contribute to the progression of cancer [8]. The regulation of mRNA has been considered as an important target for tumor development and cell fate determination [9]. HNRNP A1 is one of the HNRNPs that has been shown to modulate essentially every aspect of mRNA metabolism [10]. HNRNP A1 is involved in the maintenance of telomeres and cell differentiation, which are associated with oncogenesis [11]. Accumulating evidence has revealed that HNRNP A1 might strongly correlate with human cancer, such as lung carcinoma, gastric cancer, and colorectal cancer, which is overexpressed in these cancers compared to adjacent tissues [10,12,13,14]. However, the biological functions and molecular mechanisms of HNRNP A1 in HCC have not been exposed clearly. In this study, we found that HNRNP A1 promotes the progression of HCC both in vitro and in vivo.

The induction of tumor cell senescence is a promising direction for antitumor therapy. In a recent study, the downregulation of HNRNP A1 was found to induce the senescence of non-small cell lung cancer cells to inhibit the development of cancer [15]. The cell cycle was arrested in the G0/G1 phase in lung adenocarcinoma cells, which is characterized by cell senescence, after knocking down of HNRNP A1 [16]. Moreover, it was found that senescent liver cells possess a higher expression of the biomarkers of necroptosis [17]. There is a tendency for mitochondrial fusion in senescing cells [18], which may restrain the development of cancer [19]. However, the mechanism of HNRNP A1 inducing the senescence of hepatocellular carcinoma cells has not been clarified. Our study indicated that HNRNP A1 can promote the development of HCC by cell senescence mediated by cell cycle arrest, escaping from necroptosis and inhibiting the expression of mitochondrial fusion protein.

In this study, we aimed to investigate the role of HNRNP A1 in HCC. Our results indicated that the downregulation of the HNRNP A1 could inhibit the development and progression of HCC. These findings also provide a basis for HNRNP A1 as a potential therapeutic target for HCC.

## 2. Results

### 2.1. HNRNP A1 Is Upregulated in Human HCC Tissues and Cell Lines

To investigate the expression of HNRNP A1 between HCC tissues and normal tissues, the TNMplot database was used to analyze this difference. As shown in Figure 1A, compared with the normal tissues, the expression of HNRNP A1 is higher in human HCC tissues. We collect 10 pairs of HCC tissues and adjacent normal tissues and examined the relative expression of HNRNP A1 by qRT-PCR and Western blot. Consistent with the database results, the expression of HNRNP A1 in HCC tissues was significantly higher than that in adjacent normal tissues (Figure 1B,C). Immunohistochemical analysis revealed over-abundant expression of HNRNP A1 in HCC tissues than that in adjacent normal tissues (Figure 1D). We further examined the expression of HNRNP A1 in normal hepatocytes cell line LO2 cells and HCC cells lines (LM3, HUH7, LI7, SNU387) by qRT-PCR and Western blot. Most of the HCC cells lines, except SNU387, expressed more HNRNP A1 than LO2 (Figure 1E,F). These data indicated that HNRNP A1 was upregulated in HCC tissues.

### 2.2. HNRNP A1 Promotes the Proliferation of HCC Cell Lines In Vitro and In Vivo

To investigate the role of HNRNP A1 in HCC cell lines, the CCK-8 assay was used to detect the proliferation ability of HCC cells and LO2. The results revealed that the growth rate of different HCC cells is positively correlated with the expression of HNRNP A1 (Figure 2A). We selected LM3 and LI7 cells to verify the biological function of HNRNP A1. A lentivirus was constructed to infect LM3 cells (LM3-siHNRNP A1) and LI7 cells (LI7-siHNRNP A1), respectively. qRT-PCR and Western blot were used to verify the efficiency of transfection (Appendix A). The CCK-8 assay revealed that LM3 and LI7 cells transfected with inhibitor lentivirus possessed a lower growth rate (Figure 2A,B). Consistent with these results, the colony formation assay showed that the inhibition of HNRNP A1 could suppress HCC cell proliferation (Figure 2C,D). Furthermore, these cells were injected subcutaneously into nude mice to demonstrate the effects of HNRNP A1 on tumor growth in vivo. The LM3-siHNRNP A1 and LI7-siHNRNP A1 groups exhibited a significant decrease in tumor volume compared with the control group (Figure 2E,F). In summary, all the results mentioned above revealed that the expression of HNRNP A1 could promote the proliferation of HCC cells in vitro and in vivo.

### 2.3. HNRNP A1 Positively Regulates the Migration and Invasion of HCC Cell Line

To further investigate the promotion role of HNRNP A1 in HCC cells, matrigel invasion assays were used to examine the effects of HNRNP A1 on HCC cell invasion. We retrieved the expression of HNRNP A1 in samples of different tumor grades from the TCGA database. The expression of HNRNP A1 in well-differentiated and moderately differentiated tumors was significantly lower than that in poorly differentiated and undifferentiated tumors, which indicates that HNRNP A1 may have a role in promoting HCC invasion (Appendix A). As shown in Figure 3A,B, compared with the normal HCC cell lines, the number of invasive cells was obviously decreased by HNRNP A1 ablation in LM3 cells and LI7 cells. In the wound healing assay, the downregulation of HNRNP A1 reduced the migration rate of HCC cells (Figure 3C–E). These results suggested that HNRNP A1 is crucial to cell migration and invasion in HCC cells.

### 2.4. HNRNP A1 Inhibits the Senescence of Hepatocellular Carcinoma Cells

Previous studies have shown that HNRNP A1 plays an important role in inhibiting cell senescence [20,21]. Nevertheless, the mechanism of HNRNP A1-mediated cell senescence in HCC is not clear. In order to clarify this problem, we detected biomarkers of cell senescence between the HCC cell lines and siHNRNP A1 groups. As shown in Figure 4A, compared with LO2 cells, the HCC cell lines, LM3 and LI7, hardly express the overwhelming majority of the senescence-associated secretory phenotype (SASP). Miraculously, the knocking down of HNRNP A1 evidently upregulated the expression of most of the SASP (Figure 4B). Consistent with the result of the SASP, the mRNA expression of Ki67 also decreased significantly in LM3-siHNRNP A1 and LI7-siHNRNP A1 (Figure 4C). Moreover, the expression of P53, which has been viewed as one of the famous biomarkers of cell senescence, was upregulated after knocking down HNRNP A1 (Appendix A). Therefore, we next examined the activity of senescence-associated beta-galactosidase (SA-β-Gal) and the expression of the GLB-1 gene, which encodes SA-β-Gal (Figure 4D,E). The percentage of SA-β-Gal-positive cells was increased by HNRNP A1 knockdown in HCC cell lines. These results reveal that HNRNP A1 inhibits the senescence of hepatoma cells.

### 2.5. HNRNP A1 Regulates Cell Cycle through P16^INK4^

Cell cycle arrest is characterized by cell senescence, which leads to growth inhibition [21]. A previous study proved that HNRNP A1 can affect the stability of P16^INK4^, a kind of the SASP, and then regulate the cell cycle [22]. As our previous studies demonstrated that the mRNA expression of P16^INK4^ is significantly upregulated when HNRNP A1 is knocked down, we examined the protein level of P16^INK4^ (Figure 4F,G). The expression of P16^INK4^ is apparently increased when HNRNP A1 is knocked down. At the same time, after treatment with VPC-80051, an inhibitor of HNRNP A1, P16^INK4^ increased gradually. We observed that the cell cycle was arrested at the G0/G1 phase at this moment (Figure 4G,H). However, when P16^INK4^ was knocked out by lentivirus, the cell cycle was accelerated and tended to return to the original position. These findings revealed that HNRNP A1 promotes cell cycle by downregulating P16^INK4^ in HCC cell lines.

### 2.6. HNRNP A1 Regulates Necroptosis and Mitochondrial Dynamics

Recent research showed that cell senescence has connections with necroptosis [23]. In order to verify whether HNRNP A1 is related to necrotic apoptosis, we detected the expressions of RIPK1, RIPK3, and MLKL, viewed as biomarkers of necroptosis. Expectedly, the expressions of RIPK1, RIPK3, and MLKL were upregulated in LM3-siHNRNP A1 cells and LI7-siHNRNP A1 compared with normal HCC cell lines (Figure 5A,B). There is abundant evidence for a relationship between cell senescence and mitochondrial dynamics. Our data indicated that the expressions of MFN1 and MFN2, which contribute to mitochondrial fusion, were upregulated, and mitochondrial fission proteins drp1 and fis1 showed no obvious change (Figure 5C,D). These data showed that HNRNP A1 can help HCC cells escape from necroptosis and plays a critical role in mitochondrial dynamics.

## 3. Discussion

In this study, we found that HNRNP A1 is apparently upregulated in human HCC tissues and cell lines. On the basis of a set of experiments, the expression of HNRNP A1 promoted the proliferation, migration, and invasion of HCC in vitro and accelerated tumorigenesis in vivo, which showed that HNRNP A1 may be a potential target for HCC. Furthermore, our research also implied that HNRNP A1 also regulates the cellular senescence of HCC. This regulation is decided by the change of the cell cycle through p16^IK4a^. Moreover, HNRNP A1 also influences necroptosis and mitochondrial dynamics in HCC cell lines.

The aberrant expression of HNRNP A1 has been proven to regulate the transcription and splicing of mRNA and to be related to the development and progression of cancer [24,25,26]. Previous studies elucidated that HNRNP A1 was highly expressed in most tumors [27] and revealed the positive role of HNRNP A1 in promoting different malignancies such as lung cancer, colorectal cancer, and gastric cancer [10,12,13,14]. However, the mechanism of HNRNP A1 in HCC has not been fully elucidated. Researchers detected that the upregulation of HNRNP A1 can promote HCC metastasis through facilitating the EMT and invasion/migration of HCC cells, and the loss of HNRNP A1 leads to reduced levels of EMT markers [28,29,30,31]. In our research, HNRNP A1 played a critical role in the proliferation, invasion, and migration of HCC cell lines, where HNRNP A1 was highly expressed. We also noticed that SNU387, which has a lower expression level of HNRNP A1, also possess a faster proliferation rate than LO2. The role of HNRNP A1 at a lower level deserves further study. Moreover, HNRNP A1 plays an important role in aerobic glycolysis and promotes tumor proliferation [12,32]. Our results suggested that the repression of HCC cellular senescence may become another reason.

Cell senescence has long been considered as a tumor-protective mechanism [33,34], and to be an indicator of cancer treatment [35,36]. HNRNP A1 is reported to be involved in post-transcriptional regulation to antagonizing cellular senescence [37,38]. We tested whether HNRNP A1 can modulate cell senescence in HCC. Our results showed that HNRNP A1 enables HCC cells to escape from senescence. Furthermore, other researchers proved that several kinds of SASPs were downregulated when HNRNP A1 was overexpressed [37]. Other studies elucidated that the expression of HNRNP A1 decreased in senescent cells, and the ratio of p14^ARF^ to p16^IK4a^ was lower in senescent cells [22]. Our results may explain these conclusions. Moreover, the subsequent downregulation of p16^IK4^ accelerated cell cycle progression, which is consistent with previous results [39,40].

Our second finding was the function of HNRNP A1 in the necroptosis of HCC. Evidence has proven that biomarkers of necroptosis can be detected in the brain tissues of aging mice [41]. The activation of necroptosis in human age-related diseases make this conclusion go a step further [42]. Our research indicated that HNRNP A1 can restrain the necroptosis of HCC. Further studies should be implemented to explore the reasons behind these results.

Cell senescence is also correlated with the structure of mitochondria. Mitochondrial morphology remains dynamic due to competition between fusion and fission [43,44]. Overexpressions of MFN1 and MFN2 were found in aged skeletal muscle [45], which suggests that the morphological change of mitochondria in senescence may be related to the imbalance of fission and fusion. Our results demonstrated that cell senescence mediated by HNRNP A1 can promote the expressions of MFN1 and MFN2. However, the intrinsic mechanism behind this remains to be studied.

In summary, our data demonstrated that HNRNP A1 was upregulated in HCC. HNRNP A1 was correlated with tumor proliferation, migration, and invasion. Moreover, the expression of HNRNP A1 inhibited the senescence of HCC cells through regulating P16 expression and then accelerating the cell cycle. We also found that HNRNP A1 regulated necroptosis and mitochondrial dynamics. In general, our findings indicated that the HNRNP A1 could be a potential therapeutic target for HCC.

## 4. Materials and Methods

### 4.1. Patients and Samples

This study conformed to the Declaration of Helsinki. All of the experimental schemes were permitted by the Ethics Committee of Nanjing University. All clinical specimens belong to Drum Tower Hospital, affiliated with Nanjing University. Every patient and his/her family signed an informed consent form for specimen collection. The 30 clinical specimens were divided into liver cancer tissues and para-cancerous tissues. One part of the specimen was used for the extraction of total RNA and total protein, and the rest was embedded into wax blocks, which were used for immunohistochemical staining.

### 4.2. Cell Culture and Gene Transfection

The cell lines mentioned in this article were purchased from Cell Bank of Type Culture Collection of the Chinese Academy of Sciences (Shanghai Institute of Cell Biology, Shanghai, China). Human hepatocellular carcinoma cell lines (LM3, Huh7) were cultured in DMEM (Gibco) supplemented with 10% FBS (Gibco). Human hepatocellular carcinoma cell lines (Li7, SNU387) and the normal liver cell line LO2 were maintained in RPMI-1640 medium (Gibco) supplemented with 10% FBS (Gibco).The HNRNP A1 siRNA and control siRNA were purchased from Shanghai Genechem Co. Ltd. (Shanghai, China), According to the product requirements, virus transfection reagents were used to transfect cells.

### 4.3. Protein Extraction and Western Blot

The protein of clinical specimens and hepatocellular carcinoma cells was distilled by lysis buffer and centrifuged at 12,000× *g* rpm for 10 min. The total protein quantity was determined by the BCA method. Proteins separated by SDS-PAGE were transferred to PVDF membranes. Specific antibodies were used for incubation. The protein bands were detected using an exposure meter in the presence of an enhanced exposure liquid.

### 4.4. RNA Extraction and Quantitative RT-PCR (qRT-PCR)

The TRIzol method was used to extract the RNA of the clinical specimens and hepatocellular carcinoma cells. We added TRIzol and chloroform to the samples. The upper layer of the solution was collected, and 500 μL of isopropanol was added after centrifugation. The sediment was washed with 75% ethanol and air-dried. The RNA precipitate was dissolved in 20 μL DEPC water, and the concentration of RNA was measured.

cDNA was prepared by amplifying 500 ng of RNA by the Prime-Script™RT Master Mix. Quantitative PCR was performed using TB Green™Premix Ex Taq™ following the manufacturer’s instructions. Data were processed in Prism 8.0 (GraphPad 8.0.2(263)). The relative expression levels were normalized to β-actin.

### 4.5. Cell Proliferation Assay, Invasion Assay, and Wound Healing Assay

The Cell Counting Kit 8 (CCK-8) assay (Dojindo Laboratories, Kumamoto, Japan) and colony formation assay were used for assessing the cell proliferation ability. The ability of cell invasion was detected by the transwell invasion assay. For the wound healing assay, equal numbers of normal cells and transfected cells were plated into six-well plates and cultured in complete media corresponding to the cancer cells. When the cells grew to 100% confluence, a 200 µL sterile gun head was utilized to scratch across the surface of the well. After incubation at 37 °C of 24 h and 48 h, the scratches were observed.

### 4.6. Immunohistochemical

IHC was utilized to analyze the protein expression of HNRNP A1 in HCC tissues and para-cancerous tissues.

### 4.7. β-Galactosidase Staining

β-galactosidase staining was carried out using an aging-associated β-galactosidase staining kit (Beyotime), following the manufacturer’s instructions. After being immobilized with fixative solution for 15 min at ambient temperature, HCC cells were incubated with the β-Galactosidase Staining Solution at 37  °C in a siccative incubator without CO_2_ overnight (cell lines). Cells or slides were examined under a microscope for the development of blue color.

### 4.8. Cell Cycle Analysis

Normal HCC cells and transfected cells were gathered and immobilized with cold 70% ethanol and stored for 30 min at 4 °C, then stained with propidium iodide (PI) and RNase A solution for 30 min at 37 °C in a darkroom. The cell distribution across the cell cycle was analyzed with a FACS Vantage SE flow cytometer.

### 4.9. In Vivo Animal Study

The tumor-bearing animal model was carried out using nude mice. Normal HCC cell and transfected cell suspensions (200 μL, containing 106 cells) were separately injected in the right armpit of each male nude mouse aged 4 weeks. The tumors were measured twice per week. Euthanasia was performed after a solid tumor had taken shape for 4 weeks. The tumors were collected and measured.

### 4.10. Statistical Analysis

Data analysis was performed using the SPSS software version 16. Each experiment was carried out at least in triplicate, and all data are presented as the mean ± SD. The χ^2^-test and Student’s *t*-test were used to estimate the statistical significance with *p* < 0.05 judged to be statistically significant.

## Figures and Tables

**Figure 1 ijms-23-10209-f001:**
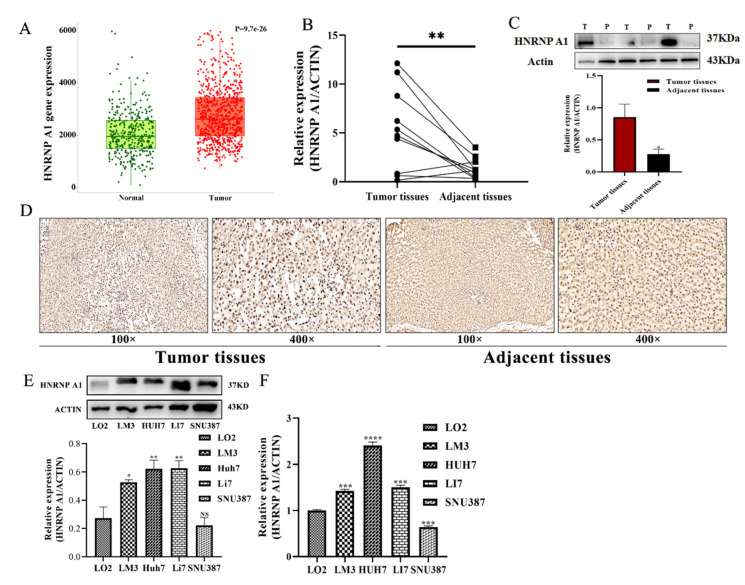
The expression of HNRNP A1 in HCC tissues, HCC cells, and the proliferation of HCC cells. (**A**) The expression levels of HNRNP A1 in HCC tissues and normal liver tissues in TNMplot; (**B**,**C**) the expression levels of HNRNP A1 in 10 pairs of human HCC tissues and adjacent normal tissues were detected using qPCR and WB; (**D**) the expression of HNRNP A1 in HCC tissues and adjacent normal tissues were detected by immunohistochemistry, *n* = 5; (**E**,**F**) the expression levels of HNRNP A1 in HCC cells and LO2 cells. * *p* < 0.05; ** *p* < 0.01; *** *p* < 0.001; **** *p* < 0.001; NS, not significant. The data are expressed as the mean ± SD.

**Figure 2 ijms-23-10209-f002:**
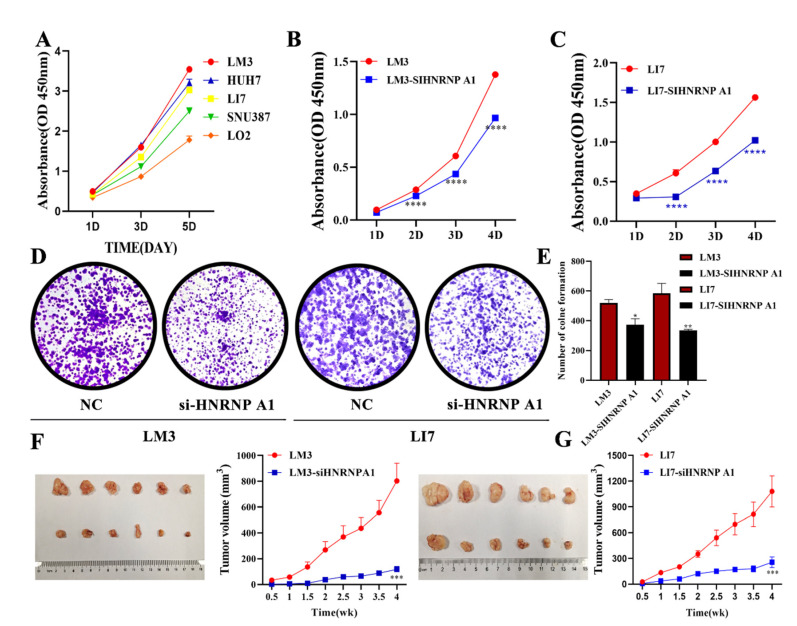
HNRNP A1 promotes the proliferation of HCC in vitro and in vivo. (**A**) The CCK-8 assay was used to explore the proliferation of HCC cells and LO2 cells; (**B**,**C**) the CCK-8 assay was used to explore the proliferation of HCC cells; (**D**,**E**) effects of HNRNP A1 expression on the colony formation of HCC cells, each picture represents the growth of the cells in one hole of a six well plate; (**F**,**G**) HCC cells were injected subcutaneously into nude mice to obtain xenograft tumors. The tumor growth curves were significantly different between HCC cells with the knockdown of HNRNP A1 and the control group.* *p* < 0.05; ** *p* < 0.01; *** *p* < 0.001; **** *p* < 0.001. The data are expressed as the mean ± SD.

**Figure 3 ijms-23-10209-f003:**
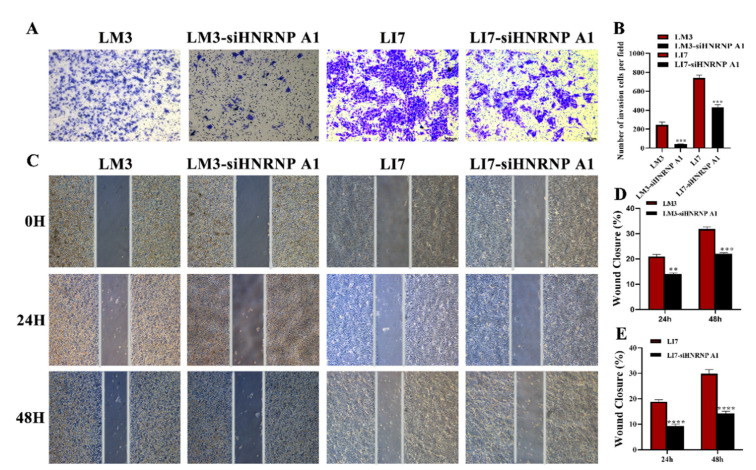
HNRNP A1 promotes the migration and invasion of HCC cells. (**A**,**B**) Matrigel invasion assays were used to examine the effects of HNRNP A1 on HCC cell invasion; the invasion capability was quantified as the cell number. Each picture was magnified 500 times by the microscope; (**C**–**E**) wound healing was performed to determine the ability of cell migration after knocking down HNRNP A1. Each picture was magnified 100 times by the microscope. ** *p* < 0.01; *** *p* < 0.001; **** *p* < 0.001. The data are expressed as the mean ± SD.

**Figure 4 ijms-23-10209-f004:**
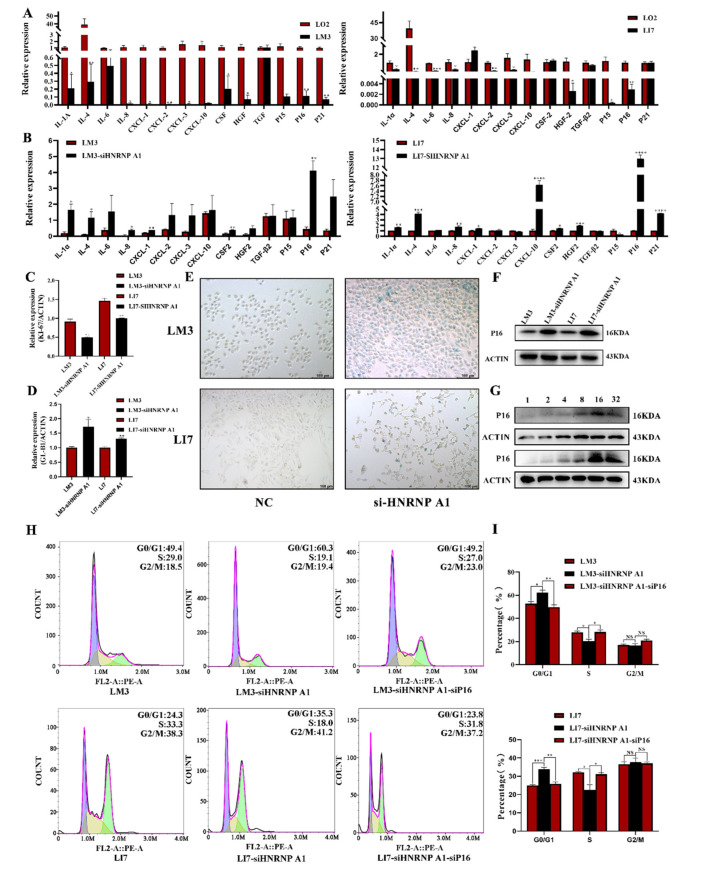
HNRNP A1 inhibits the senescence of hepatocellular carcinoma through the cells’ cell cycle mediated by P16^INK4^. (**A**) The expression levels of the SASP in HCC cell lines, LO2; (**B**) the change of the expression levels of the SASP after knocking down of HNRNP A1; (**C**) the expression of KI-67 after knocking down of HNRNP A1; (**D**,**E**) the expression of GLB-1 and representative images of SA-β-gal staining in HCC cells with the knockdown of HNRNP A1 and the control group; (**F**) the expression of P16^INK4^ after knocking down of HNRNP A1 detected by Western blot; (**G**) promotion of the levels of P16INK4 upon treatment with 8 µM of VPC-80051 in LM3 cells and 16 µM of VPC-80051 in LI7 cells; (**H**,**I**) effects of the HNRNP A1 and P16^INK4^ difference in the cell cycle distribution of HCC cells. * *p* < 0.05; ** *p* < 0.01; *** *p* < 0.001; **** *p* < 0.001. The data are expressed as the mean ± SD.

**Figure 5 ijms-23-10209-f005:**
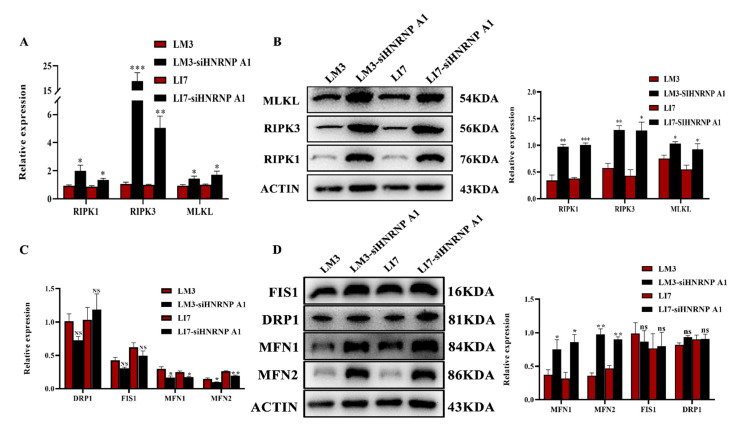
HNRNP A1 regulates necroptosis and mitochondrial dynamics. (**A**,**B**) The expression of RIPK1, RIPK3 and MLKL after knocking down of HNRNP A1 detected by qRT-PCR and WB. (**C**,**D**) The expression of DRP1, FIS1, MFN1 and MFN2 after knocking down of HNRNP A1 detected by qRT-PCR and WB. * *p* < 0.05; ** *p* < 0.01; *** *p* < 0.001; NS, not significant. The data expressed as the mean ± SD.

## Data Availability

Not applicable.

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
