# Peer review of "Tumor-Promoting Actions of HNRNP A1 in HCC Are Associated with Cell Cycle, Mitochondrial Dynamics, and Necroptosis"

_ijms, 2022, doi:10.3390/ijms231810209_

Round 1

Reviewer 1 Report

To understand the role of heterogenous nuclear ribonucleoproteins (HNRNPs) in liver cancer, the authors conducted a study and found that A1 is associated with cell cycle, mitochondrial dynamics, and necroptosis.

The experiment appears to be carried out neatly and carefully. Also, the writing of the thesis itself will be sufficient.

If possible, I would like you to consider this. At 1C, the SMU387 cell line expressed less than LO2. What happens when, for example, HNRNP A1 is introduced into this cell line?

Furthermore, in 1A, when looking at the actual cases as a whole, even though the expression is higher in the tumor tissue, a considerable number of cases overlap. So, I would like to hear reports on some of the roles of this molecule in low-income cases or cell lines.

Wouldn't that lead to extracting some kind of guideline to distinguish between cases in which this molecule has a large effect and cases in which it does not?

I can't find any other major issues.

Author Response

Dear Reviewers:

        Thank you for your letter and for the reviewers’ comments concerning our manuscript entitled “Tumor-promoting actions of HNRNP A1 in HCC is associated with cell cycle, mitochondrial dynamics and Necroptosis” (ID: ijms-1878576). Those comments are all valuable and very helpful for revising and improving our paper, as well as the important guiding significance to our researches. We have studied comments carefully and have made correction which we hope meet with approval. Revised portion are marked in red in the paper. The main corrections in the paper and the responds to the reviewer’s comments are as flowing:

Responds to the reviewer’s comments:

1.Response to comment: At 1C, the SNU387 cell line expressed less than LO2. What happens when, for example, HNRNP A1 is introduced into this cell line?

Response: Thank for your comments. As Reviewer suggested that introduction HNRNP A1 into cell line who express less HNRNP A1 will promote tumor progression. Studies had confirmed that cell proliferation was accelerated when the expression of HNRNP A1 is increased in HCC[1] and other cancer[2]. We have added these results in the discussion sector of this revision (line 162-166, page 10). Thank you so much for your careful check.

2.Response to comment: Furthermore, in 1A, when looking at the actual cases as a whole, even though the expression is higher in the tumor tissue, a considerable number of cases overlap. So, I would like to hear reports on some of the roles of this molecule in low-income cases or cell lines.

Response: We gratefully appreciate for your valuable suggestion. In order to further prove the expression of HNRNP A1 in hepatocellular carcinoma, we retrieve data from TCGA(supplementary Fig 2). It seems that the expression of HNRNP A1 in different grades is much higher than that in normal tissues. Further more, it is elucidated that HNRNPA1 was highly expressed in most tumors, except in UCEC, BLCA, and HNSC[3]. However, there is no article to be found to clarify its function in UCEC and HNSC. When it comes to BLCA, recent study showed that HNRNP A1 mediated pre-mRNA alternative splicing of the PKM gene, which still play a positive role in BLCA[4]. Thank for your comments. We have added these results in the results of this revision (line 157, page 9). Special thanks to you for your good comments.

      We tried our best to improve the manuscript and made some changes in the manuscript.  These changes will not influence the content and framework of the paper. And here we did not list the changes but marked in red in revised paper. We appreciate for Editors/Reviewers’ warm work earnestly, and hope that the correction will meet with approval.

  1. Chen, D.; Wang, Y.; Lu, R.; Jiang, X.; Chen, X.; Meng, N.; Chen, M.; Xie, S.; Yan, G.R. E3 ligase ZFP91 inhibits Hepatocellular Carcinoma Metabolism Reprogramming by regulating PKM splicing. Theranostics 2020, 10, 8558-8572, doi:10.7150/thno.44873.
  2. Ryu, H.G.; Jung, Y.; Lee, N.; Seo, J.Y.; Kim, S.W.; Lee, K.H.; Kim, D.Y.; Kim, K.T. HNRNP A1 Promotes Lung Cancer Cell Proliferation by Modulating VRK1 Translation. Int J Mol Sci 2021, 22, doi:10.3390/ijms22115506.
  3. Li, H.; Liu, J.; Shen, S.; Dai, D.; Cheng, S.; Dong, X.; Sun, L.; Guo, X. Pan-cancer analysis of alternative splicing regulator heterogeneous nuclear ribonucleoproteins (hnRNPs) family and their prognostic potential. J Cell Mol Med 2020, 24, 11111-11119, doi:10.1111/jcmm.15558.
  4. Yan, Q.; Zeng, P.; Zhou, X.; Zhao, X.; Chen, R.; Qiao, J.; Feng, L.; Zhu, Z.; Zhang, G.; Chen, C. RBMX suppresses tumorigenicity and progression of bladder cancer by interacting with the hnRNP A1 protein to regulate PKM alternative splicing. Oncogene 2021, 40, 2635-2650, doi:10.1038/s41388-021-01666-z.

Reviewer 2 Report

This manuscript by Zhao et al evaluates the potential role of HNRNP A1 in the development and progression of Hepatocellular carcinoma (HCC). HNRNP A1 was found to be upregulated in human HCC tissues and cell lines. HNRNP A1 was also found to promote cell proliferation in vitro and in vivo and promote the migration and invasion of HCC cell lines. Mechanistically, HNRNP A1 inhibits senescence and necroptosis in HCC cells by regulating the levels of p16INK4­.

Overall, this is a well-designed study which will be helpful to identify novel strategies for targeting HCC. I here provide a few comments to help strengthen the findings of the study.

Major comments:

1.      Does overexpression of HNRNP A1 in low expressing cell lines like LO2 increase cell proliferation?

2.      Authors write that “qRT-PCR and western blot was used to verify the efficiency of transfection”. These results must be included in the manuscript.

3.      Does loss of HNRNP A1 lead to reduced levels of EMT markers?

4.      Is there a positive correlation between tumor grade and invasion and HNRNP A1 levels? The authors should include this in Figure 3.

5.      The levels of p21 are unchanged between LO2 and L17 cells, despite reduced senescence in L17 cells. How do the authors explain this?

6.      Does overexpression of HNRNP A1 cause a reduction in SASP markers?

7.      The authors should perform immunohistochemistry to test the levels of p16INK4 in tumors with high HNRNP A1 vs low HNRNP A1.

8.      Is the effect of HNRNP A1 loss on necroptosis direct or an indirect consequence of increased cell senescence? To clarify this, the authors should test if co-depletion of p16INK4 in cells lacking HNRNP A1 can rescue the levels of necroptosis markers.

9.      Is the overexpression of HNRNP A1 dependent on p53 status? What is the status of p53 in the cell lines tested?

Minor comments:

1.      For better flow, Figure 1G could be moved to Figure 2.

Author Response

Dear Reviewers:

Thank you for your letter and for the reviewers’ comments concerning our manuscript entitled “Tumor-promoting actions of HNRNP A1 in HCC is associated with cell cycle, mitochondrial dynamics and Necroptosis” (ID: ijms-1878576). Those comments are all valuable and very helpful for revising and improving our paper, as well as the important guiding significance to our researches. We have studied comments carefully and have made correction which we hope meet with approval. Revised portion are marked in red in the paper. The main corrections in the paper and the responds to the reviewer’s comments are as flowing:

Responds to the reviewer’s comments:

  1. Response to comment: Does overexpression of HNRNP A1 in low expressing cell lines like LO2 increase cell proliferation?

Response: Thank for your comments. It is truly as reviewer suggested that HNRNP A1 promote cell proliferation when upregulated. It is elucidated that HNRNPA1 was highly expressed in most tumors, except in UCEC[1]. However, there is no article to be found to clarify its function in UCEC. Moreover, studies had confirmed that cell proliferation was accelerated when the expression of HNRNP A1 is increased in HCC[2] and other cancer[3]. We have added these results in the discussion sector of this revision (line 162-165, page 10). Thank you so much for your careful check.

  1. Response to comment: Authors write that “qRT-PCR and western blot was used to verify the efficiency of transfection”. These results must be included in the manuscript.

Response: We appreciate the reviewers very much for point out this important issue. We are very sorry for our negligence of this problem. We have made correction according to the reviewer’s comments. This part will be presented in the Supplementary Fig 1.

  1. Response to comment: Does loss of HNRNP A1 lead to reduced levels of EMT markers?

Response: This is a constructive suggestion by the reviewers. We have added these contents in the discussion sector of this revision, which will help readers to better understand the effect of HNRNP A1(line 162, page 10). We noticed that in previous studies, the effect of HNRNP A1 expression level on EMT has been confirmed in some cancers. Downregulation of HNRNP A1 can upregulate the expressions of epithelial marker E-cadherin and tight junction protein ZO-1, and downregulate the expressions of mesenchymal markers N-cadherin, vimentin, Twist 1 and Nanog in HCC cell lines, whereas HNRNP A1 overexpression abolished these effects[4]. The researchers also observed this phenomenon in HEK-293 cells and HeLa cells[5].

  1. Response to comment: Is there a positive correlation between tumor grade and invasion and HNRNP A1 levels? The authors should include this in Figure 3.

Response: We gratefully appreciate for your valuable suggestion. Because the quantity of samples studied in this manuscript is far from enough to analyze reliable results, we retrieve data from TCGA(Supplementary Fig 2). It seems that the expression of HNRNP A1 in well differentiated and moderately tumors was significantly lower than that in poorly differentiated and undifferentiated tumors. These data will be a strong evidence that HNRNP A1 promotes HCC invasion. Thank for your comments. We have added these results in the results of this revision.

  1. Response to comment: The levels of p21 are unchanged between LO2 and L17 cells, despite reduced senescence in L17 cells. How do the authors explain this?

Response: Thank you for pointing out this problem in manuscript. We are very sorry for our vague writing. What we want to deliver is that most of the expression of SASP in HCC cells is absent or very low compared with LO2, a kind of hepatocyte. However, when it comes to the absent of HNRNP A1, some of the SASP rise sharply. We due this phenomenon to the senescence of cells, because studies have found the upregulate of P21 when hepatocyte was induced senescence [6,7]. We have re-written this part according to the Reviewer’s suggestion. (line 112, page 6)

  1. Response to comment: Does overexpression of HNRNP A1 cause a reduction in SASP markers?

Response: We gratefully appreciate for your valuable suggestion. SASP is a set of proteins, and we selected several of them and observed the change when HNRNP A1 was downregulated. Previous studies have proved that IL-6 and IL-8 was downregulated when HNRNP A1 is overexpressed[8].

But other kind of SASP needs to be confirmed by subsequent experiments. We have added these contents in the discussion sector of this revision (line 174, page 9).

  1. Response to comment: The authors should perform immunohistochemistry to test the levels of p16INK4 in tumors with high HNRNP A1 vs low HNRNP A1.

Response: This is a constructive suggestion by the reviewers. We totally understand the reviewer's concern. According to the current results, tissues highly expressing HNRNP A1 should express less p16INK4 and vice versa. We will be happy to carry out this experiment if we have enough time. However, due to the rebound of COVID-19 epidemic in China, it takes a long time to purchase materials to carry out this experiment. We will further verify this result in the next stage of experiments. Thank you so much for your careful check.

  1. Response to comment: Is the effect of HNRNP A1 loss on necroptosis direct or an indirect consequence of increased cell senescence? To clarify this, the authors should test if co-depletion of p16INK4 in cells lacking HNRNP A1 can rescue the levels of necroptosis markers.

Response: We gratefully thanks for the precious time the reviewer spent making constructive remarks. It is confessedly that cell senescence has connections with necroptosis[9]. In our manuscript, we found that the changes of HNRNP A1 could affect cell senescence and necroptosis. However, there is currently no direct evidence on the relationship between them. This will be our key research objective in the next stage. Reviewer's suggestion gave us great inspiration about that. Special thanks to you for your good comments.

  1. Response to comment: Is the overexpression of HNRNP A1 dependent on p53 status? What is the status of p53 in the cell lines tested?

Response: We gratefully appreciate for your valuable suggestion. P53 is one kind of famous biomarkers of cell senescence. We detected P53 in normal HCC cell line and cells whose HNRNP A1 was knocked down. The experimental results shown that P53 was activated after HNRNP A1 knocking down. We have added these results in the results of this revision (Supplementary Fig 3) (line 116-117, page 6).

  1. Response to minor comments: For better flow, Figure 1G could be moved to Figure 2.

Response: Thank you for your rigorous consideration. We have changed the picture correction according to the reviewer’s comments. Special thanks to you for your good comments.

We tried our best to improve the manuscript and made some changes in it.  These changes will not influence the content and framework of the paper. And here we did not list the changes but marked in red in revised paper. We appreciate for reviewers’ warm work earnestly, and hope that the correction will meet with approval. Once again, thank you very much for your comments and suggestions.

  1. Li, H.; Liu, J.; Shen, S.; Dai, D.; Cheng, S.; Dong, X.; Sun, L.; Guo, X. Pan-cancer analysis of alternative splicing regulator heterogeneous nuclear ribonucleoproteins (hnRNPs) family and their prognostic potential. J Cell Mol Med 2020, 24, 11111-11119, doi:10.1111/jcmm.15558.
  2. Chen, D.; Wang, Y.; Lu, R.; Jiang, X.; Chen, X.; Meng, N.; Chen, M.; Xie, S.; Yan, G.R. E3 ligase ZFP91 inhibits Hepatocellular Carcinoma Metabolism Reprogramming by regulating PKM splicing. Theranostics 2020, 10, 8558-8572, doi:10.7150/thno.44873.
  3. Ryu, H.G.; Jung, Y.; Lee, N.; Seo, J.Y.; Kim, S.W.; Lee, K.H.; Kim, D.Y.; Kim, K.T. HNRNP A1 Promotes Lung Cancer Cell Proliferation by Modulating VRK1 Translation. Int J Mol Sci 2021, 22, doi:10.3390/ijms22115506.
  4. Wen, Z.; Lian, L.; Ding, H.; Hu, Y.; Xiao, Z.; Xiong, K.; Yang, Q. LncRNA ANCR promotes hepatocellular carcinoma metastasis through upregulating HNRNPA1 expression. RNA Biol 2020, 17, 381-394, doi:10.1080/15476286.2019.1708547.
  5. Bonomi, S.; di Matteo, A.; Buratti, E.; Cabianca, D.S.; Baralle, F.E.; Ghigna, C.; Biamonti, G. HnRNP A1 controls a splicing regulatory circuit promoting mesenchymal-to-epithelial transition. Nucleic Acids Res 2013, 41, 8665-8679, doi:10.1093/nar/gkt579.
  6. Duan, J.L.; Ruan, B.; Song, P.; Fang, Z.Q.; Yue, Z.S.; Liu, J.J.; Dou, G.R.; Han, H.; Wang, L. Shear stress-induced cellular senescence blunts liver regeneration through Notch-sirtuin 1-P21/P16 axis. Hepatology 2022, 75, 584-599, doi:10.1002/hep.32209.
  7. Wang, M.J.; Chen, J.J.; Song, S.H.; Su, J.; Zhao, L.H.; Liu, Q.G.; Yang, T.; Chen, Z.; Liu, C.; Fu, Z.R.; et al. Inhibition of SIRT1 Limits Self-Renewal and Oncogenesis by Inducing Senescence of Liver Cancer Stem Cells. J Hepatocell Carcinoma 2021, 8, 685-699, doi:10.2147/jhc.S296234.
  8. Wang, H.; Han, L.; Zhao, G.; Shen, H.; Wang, P.; Sun, Z.; Xu, C.; Su, Y.; Li, G.; Tong, T.; et al. hnRNP A1 antagonizes cellular senescence and senescence-associated secretory phenotype via regulation of SIRT1 mRNA stability. Aging Cell 2016, 15, 1063-1073, doi:10.1111/acel.12511.
  9. Li, D.; Meng, L.; Xu, T.; Su, Y.; Liu, X.; Zhang, Z.; Wang, X. RIPK1-RIPK3-MLKL-dependent necrosis promotes the aging of mouse male reproductive system. Elife 2017, 6, doi:10.7554/eLife.27692.

Round 2

Reviewer 1 Report

Authors answered and modified their manuscript according to the reviewers' comments.

Reviewer 2 Report

The authors have addressed all the concerns. In my view, the revised manuscript is fit for publication.